# DEEP PDE SOLVERS FOR SUBGRID MODELLING AND OUT-OF-DISTRIBUTION GENERALIZATION

## ABSTRACT

Climate and weather modelling (CWM) is an important area where ML models are used for subgrid modelling: making predictions of processes occurring at scales too small to be resolved by standard solution methods. These models are expected to make accurate predictions, even on out-of-distribution data, and are additionally constrained to respect important physical constraints of the ground truth model. While many specialized ML PDE solvers have been developed, the particular requirements of CWM models have not been addressed so far. The goal of this work is to address them. We propose and develop a novel architecture, which matches or exceeds the performance of standard ML models, and which demonstrably succeeds in OOD generalization. The architecture is based on expert knowledge of the structure of PDE solution operators, which permits the model to also obey important physical constraints.

## 1 INTRODUCTION

Climate and weather modelling (CWM) is an important area which puts particular demands on machine learning (Kashinath et al., 2021). Traditional climate and weather models break the ocean, atmosphere, and land up into many grid points in order to predict future climate conditions (Brasseur & Jacob, 2017). CWM processes are represented by time-dependent partial differential equations of fluid mechanics (Mcsweeney & Hausfather, 2018). Features that are too small or complex to be explicitly calculated in the model are approximated using coarser grids (Balaji et al., 2022).

Recently, ML approaches have been used to make better approximations of these subgrid processes (Weyn et al., 2019) (Bretherton et al., 2022), (Watt-Meyer et al., 2021). For example, Bolton & Zanna (2019) applied deep learning to ocean modelling, and found that they could decrease the data resolution by a factor of 5-10 while maintaining accuracy and conservation of momentum. However, these models fail to generalize to out-of-distribution (OOD) data and they can violate physical constraints (Kashinath et al., 2021), two requirements of the CWM models.

In this work, we propose a tractable model problem which captures key aspects of the CWM problem: learning subgrid PDE solvers from sampled data, which generalize to new data distributions and satisfy physical constraints. We deliberately limit our scope to allow us to focus on these challenges from both a practical and a theoretical point of view. We are able to make definite progress in all these areas, which is a starting point for further study. Physical constraints are maintained by incorporating them directly into our model's architecture and the model is also flexible to allow for varying subgrid resolutions, ensuring accuracy even as the resolution is decreased. Moreover, we demonstrate that the PDE-inspired architecture generalizes to OOD data (while off-the-shelf neural networks do not). As the scientific community continues to grapple with complex physical phenomena, the methodologies and insights presented in this paper could serve as foundational pillars for the next generation of modelling tools in diverse physical sciences domains.

Our key contributions are:

1. **OOD Generalization for PDE Solution Operators:** By restricting to a principled architecture grounded in theory, we show that we can accurately approximate the true solution operator, even on ODD data.

2. **Accurate subgrid models:** Addressing the needs of scientific computing, we develop and integrate subgrid solvers into our model which maintain accuracy even at reduced grid resolutions (data dimensionality).

3. **Physical Constraints satisfied:** Our model hypothesis class has the benefits of traditional PDE solvers, which satisfy physical constraints, in the framework of a neural network training pipeline.

The main results are presented in the following figures, which are discussed in more detail later.

Figure 1 shows the results of the experiments for the fully resolved grid and for different subgrid models in both one and two variables. Figure 2 shows an example of a two-dimensional subgrid problem with resolution 64 ($8 \times 8$). Figure 3 shows the same phenomenon in one dimension, in a subgrid problem with a resolution of 32.

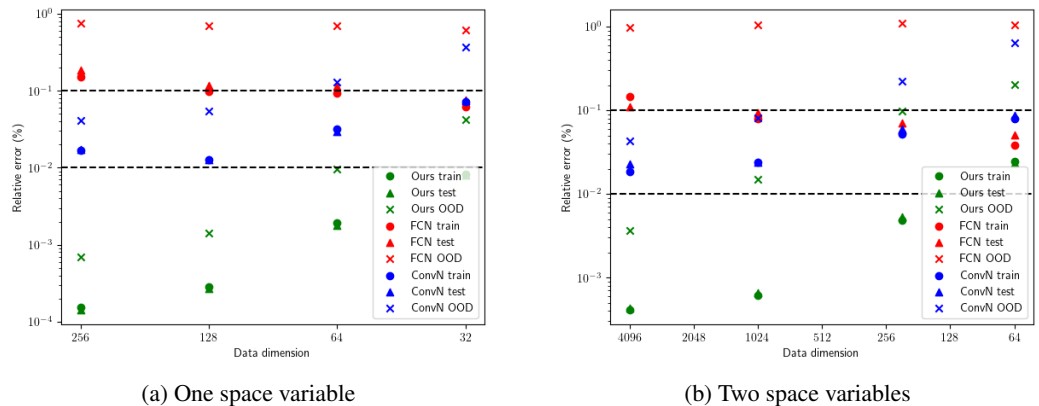

(a) One space variable         (b) Two space variables

Figure 1: In-distribution and out-of-distribution relative errors for subgrid models in one and two dimensions

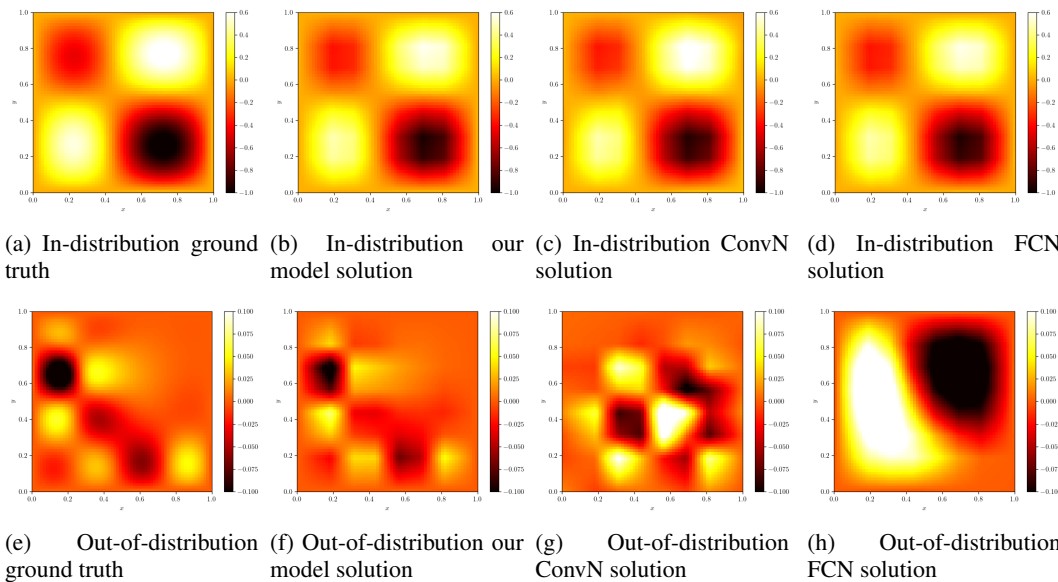

(a) In-distribution ground truth

(b) In-distribution our model solution

(c) In-distribution ConvN solution

(d) In-distribution FCN solution

(e) Out-of-distribution ground truth

(f) Out-of-distribution our model solution

(g) Out-of-distribution ConvN solution

(h) Out-of-distribution FCN solution

Figure 2: Two dimensional modelled solutions for an in-distribution and out-of-distribution example in a subgrid with a resolution of $8 \times 8$.

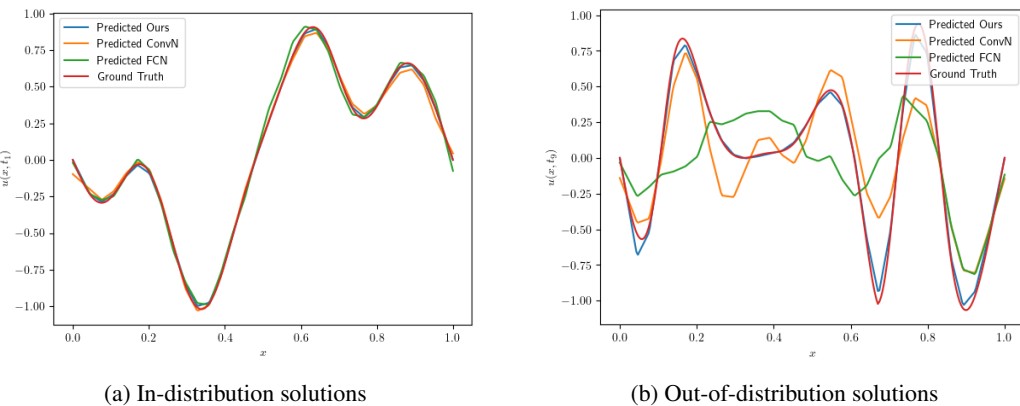

(a) In-distribution solutions                           (b) Out-of-distribution solutions

Figure 3: One dimensional modelled solutions for an in-distribution and out-of-distribution example in a subgrid with a resolution of 32.

## 2 RELATED WORK

Liu et al. (2022) build neural network models which integrate PDE operators directly in the model's architecture, while retaining the large capacity neural network architecture. They solve a different problem: learning the solution operator of a number of different PDEs but with constant coefficients. Long et al. (2018) explored the learning of coefficients for the solution operator, though they did not delve into the subgrid aspects. On the other hand, the potential issues of out-of-distribution (OOD) generalization with neural networks, especially for data with varied spectra, were highlighted by Rahaman et al. (2019). Early attempts at using physics-informed neural networks (PINNs) as PDE solvers were presented by Karniadakis et al. (2021) and Shin et al. (2020). While innovative, these PINNs occasionally struggled with accurately representing the solution operator and ensuring physical constraints.

Li et al. (2020) introduced the Fourier neural operator, which supports varying grids. However, their focus diverged towards a different PDE challenge: learning the map from the coefficients to the solution, which is different from our case, where we want to allow for different initial data to evolve in time. Recent contributions from Pfaff et al. (2021) and Han et al. (2022) present a PDE solver on irregular meshes.

The inverse problem in PDEs is to learn the coefficients of an operator, given input-output pairs (Stuart, 2010)(Kaipio & Somersalo, 2006), but does not address the subgrid aspects of a solver. Homogenization of PDEs takes the extreme approach of replacing an operator with a spatially homogeneous one (Marchenko & Khruslov, 2008), an approach which is valid in fields like material science, but not in weather and climate modelling, where the emphasis is on the heterogeneous nature of operators.

Several works connect neural network architectures and solution operators for differential equations. Chen et al. (2018) proposed neural network architecture based on ODE solvers and Haber & Ruthotto (2017) focused on the stability aspects of the architecture. Ruthotto & Haber (2020) advanced in this domain and proposed architectures based on discretized PDE solvers.

## 3 PROBLEM SETUP

### 3.1 THE PDE PROBLEM

Our PDE problem involves the function $u(\boldsymbol{x}, t)$ that satisfies:
$$\partial_t u(\boldsymbol{x}, t) = \mathcal{L}(u(\boldsymbol{x}, t); a(\boldsymbol{x}, t)), \qquad u(\boldsymbol{x}, 0) = u_0(\boldsymbol{x})$$

where $\mathcal{L}$ is the differential operator defined by the PDE, $u(\boldsymbol{x}, t)$ is the solution we seek, $a(\boldsymbol{x}, t)$ are the coefficients of the PDE (as well as the boundary conditions), and $u_0(\boldsymbol{x})$ represents the initial condition at time $t = 0$.

In this paper, we focus on the model problem of the heat equation with non-constant coefficients: $\partial_t u(\boldsymbol{x}, t) = a(\boldsymbol{x}) \Delta u(\boldsymbol{x}, t)$. This equation is chosen because it is much simpler to analyze than a system of advection-diffusion PDEs, or the Navier-Stokes equations, yet complex enough to highlight the results.

## 3.2 ML PROBLEM DEFINITION

Given samples of solutions of a time-dependent PDE on a fine grid in space, at several time slices, our goal is to learn a family of approximate solution operators. Each solution operator is to be defined for data on successively coarser grids.

We assume that the functions are all solutions of some time-dependent advection-diffusion partial differential equation, with unknown coefficients, $a(\boldsymbol{x}, t)$,

$$\partial_t u(\boldsymbol{x}, t) = L(a(\boldsymbol{x}, t), \nabla_{\boldsymbol{x}} u(\boldsymbol{x}, t), \nabla^2_{\boldsymbol{xx}} u(\boldsymbol{x}, t)), \quad u(\boldsymbol{x}, 0) = u_0(\boldsymbol{x})$$

along with some known boundary conditions.

We start with a dataset consisting of sample values of $m$ functions of the form $u_i(\boldsymbol{x}, t)$ for $\boldsymbol{x}$ in a physical domain, and $t \in [0, T]$. The functions are sampled at points $\boldsymbol{x}_j$ in a uniform grid in space of resolution $N_{\boldsymbol{x}}$, and at time intervals $T^{fine}$, consisting of $N_T$, uniformly spaced time intervals. Each function is represented by a vector of the form $U_{i,j,k} = u_i(\boldsymbol{x}_j, t_k)$ and our training dataset, on the fine grid, is of the form,

$$D^{fine} = \{ U_{i,j,k} = u_i(\boldsymbol{x}_j, t_k) \quad \boldsymbol{x}, t \in G^{fine} \times T^{fine}, i = 1, \dots, m \}$$

We are given a list of target subgrid resolutions (for example, from fully resolved to an 8 times smaller grid resolution) and coarsened data of the form,

$$D^{coarse} = \{ U_{i,j,k} = u_i(\boldsymbol{x}_j, t_k) \quad \boldsymbol{x}, t \in G^{coarse} \times T^{coarse}, i = 1, \dots, m \}$$

The goal is to learn, from the fine grid data, a solution map for each of the target grids

$$F_S(u(\boldsymbol{x}, 0); \theta) = (u(\boldsymbol{x}, t_k)), \quad \boldsymbol{x}, t \in G^{coarse} \times T^{coarse} \tag{1}$$

where $\theta$ are the parameters of the model and $T^{coarse}$ is the set of times at which we have observations on the coarse grid. By construction, we can also output the solution at other times as needed. In practice, we will use only one of the coarse grids.

The function values on the fine (well-resolved) grid would be sufficient to solve the PDE with acceptable accuracy using standard numerical PDE methods if the coefficients were known. In fact, this is how we generate the data for our dataset. However, building a solution operator on the coarse grid requires machine learning tools, since there are no analytical formulas for the operator of a coarse grid. For example, in current climate models, simplified operators are approximated, but this leads to a known loss of accuracy.

Thus, our goal is to learn a subgrid solution operator as defined in equation 1 that accurately approximates the ground truth solution, as represented by the fine grid PDE solver, in our example, or by assimilated data in a full-scale weather or climate model.

## 3.3 DATASET GENERATION

The functions $u(\boldsymbol{x}, 0)$ are generated samples with a prescribed Fourier Spectrum. We then obtain the functions $u(\boldsymbol{x}, t)$ by solving the PDE numerically on the fine grid. To obtain the coarse data for the subgrid problems, we average our data in space and sample it in time according to the stability constraints which are described in the next sections.

In our experiments, we measure the normalized $L^2$ error of the model on the training data, test data sampled from the same distribution, and on OOD data. The OOD data is obtained by generating initial conditions using a different Fourier spectrum (see figure 4), and then applying the same PDE solver to the data.

Thus, here we consider OOD to be initial data with a different shape (Fourier spectrum) from data previously seen by the models. This corresponds to the problem of having the same physical dynamics, but a different distribution of the density of particles (e.g. a more oscillatory density profile). However, we assume that the solution operator (coefficients) are the same. A different OOD problem which we do not address, would be where the underlying dynamics changed, resulting in different coefficients, which corresponds to learning a different solution operator.

## 4 OUR MODEL

Our strategy is grounded in four theoretically desirable properties for our solution operator: locality, stability, linearity, and memory-less recurrence. We implement each of those four properties into our model's architecture as follows:

**Locality:** PDEs are local operators since they depend on the derivatives of the function. Based on this, we aim to integrate the same locality property into our model architecture. To achieve this, we structure each layer as a convolutional layer which will ensure that output values are only affected by nearby input values. For all grid resolutions, we require that the solution operator is the discretization of some coarser heat equation. For this reason, it is more restrictive than a standard convolutional neural network. The convolution kernel is a diagonal multiple (corresponding to the unknown coarsened coefficients) of the fixed Laplacian operator. The convolution kernel corresponds to

$$W_{Lap,1} = \frac{dt}{dx^2}[1, -2, 1], \qquad W_{Lap,2} = \frac{dt}{dx^2} \begin{bmatrix} 0 & 1 & 0 \\ 1 & -4 & 1 \\ 0 & 1 & 0 \end{bmatrix}.$$

in one and two dimensions, respectively.

**Stability**: When solving any PDE numerically, we are bound by some stability constraints that are necessary for obtaining a convergent solution. For the heat equation, assuming we take space intervals of $dx$ (and equal in all dimensions) and time intervals of $dt$, we are bound by the stability constraint $0 \leq a(\boldsymbol{x}) \cdot \frac{dt}{dx^2} \leq \frac{1}{2 \cdot D}$ where $D$ is the dimension of the data, (Courant et al., 1967). Thus when one knows the coefficients $a(\boldsymbol{x})$ then one can simply pick $dt$ and $dx$ to satisfy the stability constraint.

In this case, we take the opposite approach. Given fixed values of $dx$ and $dt$, we can bound the coefficients themselves by

$$0 \leq a(\boldsymbol{x}) \leq C_a = \frac{dx^2}{2D \cdot dt} \tag{2}$$

This is a crucial constraint since the parameters of our model will take the place of the coefficients of the equation in our model. In this way, we design our model precisely with the aim of learning the physical process that is trying to approximate.

In order to satisfy the stability constraint, we bound the raw parameters learned by the model with a scaled sigmoid function. This is, if the model's parameters are $\theta$, then the values that we multiply with the output of the convolution layer are given by $C_a \cdot \sigma(\theta)$. This ensures that the parameters are bounded by the stability region of the PDE and thus forces the model to find a solution in the parameter space in which the PDE itself is stable.

It is important to note that when coarsening our data to subgrids, the same stability constraint must be satisfied. Thus we will always coarsen our data according to the same $\frac{dt}{dx^2}$ factor. More precisely, if we coarsen our data in space by a factor of $\lambda_x$, we will sample our time steps at intervals of $\lambda_t = \lambda_x^2$.

**Linearity:** Since the differential operator $\partial_t(\boldsymbol{x}, t) - \Delta u(\boldsymbol{x}, t)$ is linear, we want our model to be linear in the data as well. This is achieved by requiring that our model be linear in $U$, which is not typically the case for neural networks. However, the model is nonlinear in the parameters $\theta$.

**Memory-less:** Our differential operator is time-independent, meaning that no matter what the starting time $t_0$ is, the physical process is the same. Naturally, we implement this property into our

model by making each layer identical, ensuring the same physical process between each predicted time step.

Putting all of it together, our model is then a composition $f_\theta(U_0) = l_0 \circ l_0 \circ \cdots \circ l_0(U_0)$ of repeated layers. The layer is defined as (i) the convolution of the data with the fixed (non-learnable) dimension-dependent Laplacian $W_{\text{Lap, dim}}$ defined above, followed by (ii) component-wise multiplication by weights bounded between zero and a fixed, given upper bound (determined by the PDE operator as explained in equation 2), and finally (iii) this update is added back to the input vector $\boldsymbol{x}$. We note that since the bound on the weights is achieved using a sigmoid nonlinearity, the model is linear in $\boldsymbol{x}$, and nonlinear in the model parameters $\theta$.

$$l_0(U): \quad U \longrightarrow (\text{diag}(C_a \cdot \sigma(\theta))\text{conv}(W_{\text{Lap,dim}}, U)) + U \qquad (3)$$

Thus, the number of weights in the model is on the order of the number of grid points (spatial data points) as shown in tables 1 and 2. The architecture is motivated by domain expertise: the coarsened solution of a time-independent PDE should be captured approximately by an operator which also looks like a coarse solution operator (Pavliotis & Stuart, 2008). In the case where the PDE coefficients depend on time, we would have a similar structure, but with different weights for each layer.

| Subgrid resolution | 256 | 128 | 64 | 32 |
|---|---|---|---|---|
| Parameters in our model | 256 | 128 | 64 | 32 |
| Parameters in FCN | $166,720$ | $83,520$ | $41,920$ | $21,120$ |
| Parameters in ConvN | $1,130$ | $1,130$ | $1,130$ | $1,130$ |

Table 1: Model parameters for our model and the standard neural networks in one dimension

| Subgrid resolution | $4,096$ | $1,024$ | 256 | 64 |
|---|---|---|---|---|
| Parameters in our model | $4,096$ | $1,024$ | 256 | 64 |
| Parameters in FCN | $2,662,720$ | $665,920$ | $166,720$ | $41,920$ |
| Parameters in ConvN | $3,050$ | $3,050$ | $3,050$ | $3,050$ |

Table 2: Model parameters for our model and the standard neural networks in two dimensions

## 5 EXPERIMENTS

### 5.1 BASELINE NEURAL NETWORK MODEL

We conduct the experiments for both our proposed model architecture and for two baseline models which are: (1) a standard fully connected 2-layer ReLU neural network (FCN), and (2) a standard convolutional 2-layer ReLU neural network (ConvN). We chose these models given their simplicity and as a proxy for off-the-shelf ML models. The fully connected network is a simple 2-layer multilayer perceptron with a hidden layer of size 32 and ReLU activation, while the convolutional network is a simple 2-layer convolutional neural network with $3 \times 3$ kernel, ReLU activation, and hidden layer with 16 channels.

### 5.2 SUBGRID PROBLEMS

We sample initial conditions $u(\boldsymbol{x}, 0)$ from a distribution $\rho_{train}$ based on a given Fourier spectrum and then solve for the solution $u(\boldsymbol{x}, t)$ numerically for $t \leq T$ with appropriate choices of $d\boldsymbol{x}$ and $dt$ that guarantee stability. For the fully resolved grid, we simply train our model with the data generated. This is, the initial conditions are our inputs, and the solutions at the first $k$ time steps are our outputs. For the one-dimensional results presented, the fully resolved grid has size $N_{\boldsymbol{x}} = 256$, and for the two-dimensional case we have $N_{\boldsymbol{x}} = 64^2$. For both cases, we sample $k = 10$ time steps and chose $T$ large enough so that we can sample the same $k = 10$ time steps at the larger time intervals required for the subgrid models ($T = 0.002$ and $T = 0.0156$ are sufficient for the one and two-dimensional experiments carried out respectively).

For the subgrid problems, we take our data and average it down in space by a factor of $\lambda_x$ (in each dimension) and sample it down in time by a factor of $\lambda_t = \lambda_x^2$ (according to the stability conditions from the previous section). This is, the subgrid data has a dimension of $\frac{N_x}{\lambda_x^D}$ where $D$ is the number of space variables and every time step is $\lambda_t \cdot dt$ apart where $dt$ is the original, fine grid time interval.

To test out-of-distribution generalization we generate a different set of data based on a different Fourier spectrum and we sample our out-of-distribution initial conditions $\tilde{u}(\boldsymbol{x}, 0)$ from this new distribution $\rho_{ood}$. We apply the same subgrid coarsening described above and test both our model and the standard neural networks on the OOD dataset. Figure 4 shows the Fourier spectra for in-distribution and out-of-distribution data.

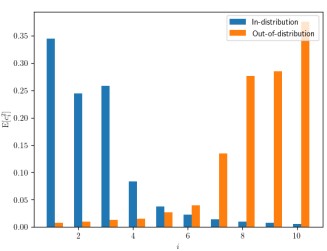 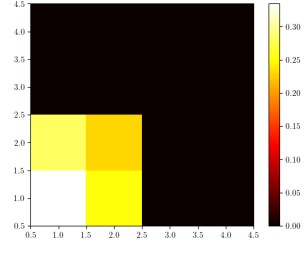 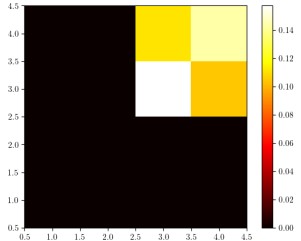

(a) Distribution spectrums for one-dimensional data

(b) In-distribution spectrum for two-dimensional data

(c) Out-of-distribution spectrum for two-dimensional data

Figure 4: Fourier Spectra for in-distribution and out-of-distribution data in one and two dimensions

We measure the error as the relative $L^2$ error with respect to the solutions, using a normalization which sets the variance of the initial data (as a function of $x$) to be one. From figure 1 we can see that the FCN achieves $10\%$ training and test error, nearly constant across grid resolutions. The error decreases slightly as the number of parameters in the model decreases, which suggests some overfitting. However, the model fails on OOD data, with a relative error close to $100\%$.

On the other hand, our model maintains high accuracy, with less than $1\%$ training error, and less than $1\%$ test error, except on the coarsest grid in two dimensions. On out-of-distribution data, the model is also quite accurate, below $10\%$ error on all subgrid problems except the coarsest grid in two dimensions which is just slightly higher. Thus we have a $10$ times improvement in distribution versus the FCN and success versus failure on out-of-distribution data. As for the ConvN, we observe that it performs significantly better than the FCN in both in-distribution and out-of-distribution data, but it still underperforms significantly compared to our model.

## 5.3 DATA COMPLEXITY

We performed an ablation study on the data complexity used in the model. We found that the fully connected neural network lost significant accuracy when exposed to data with a more complex distribution. The convolutional network lost some accuracy while our model was the most resilient to the change in complexity of the data.

Figure 6 shows the Fourier spectrum for the in-distribution and out-of-distribution data used in the study, which shows a jump in data complexity compared to the spectra used for the main results in figure 4. Figure 5 shows that the fully connected neural network was unable to learn an accurate solution operator when trained with data from the complex Fourier spectrum, with errors around $50\%$. On the other hand, our model exhibited a stable pattern across both sets of data, demonstrating that it is resilient to changes in the data complexity. The convolutional network stood in the middle, losing some accuracy but performing significantly better than the fully connected network. We note that at this level of data complexity, it was not possible to resolve the data at the coarsest resolution, so we stopped at $256$.

## 5.4 MODELLED SOLUTIONS

Figures 2 and 3 show an instance of the predicted solutions for both our model and the standard neural networks for both in-distribution and out-of-distribution examples. Figure 2 shows an exam-

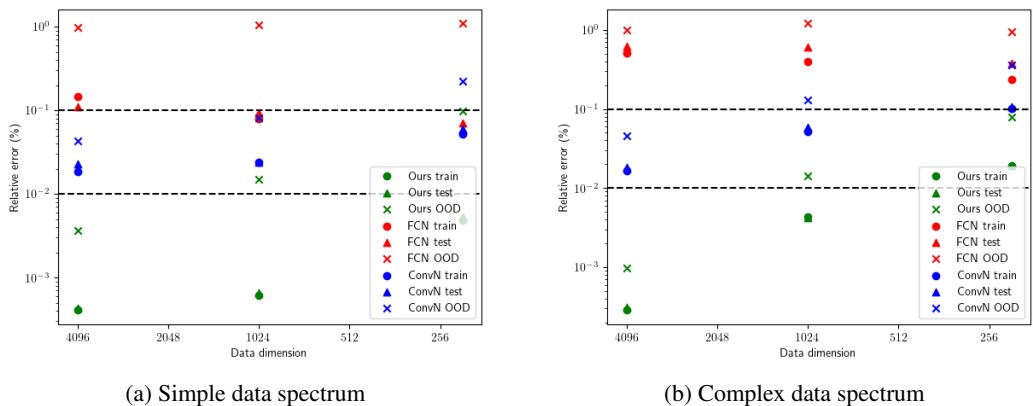

(a) Simple data spectrum        (b) Complex data spectrum

Figure 5: Subgrid errors for both simple Fourier spectra and complex Fourier spectra

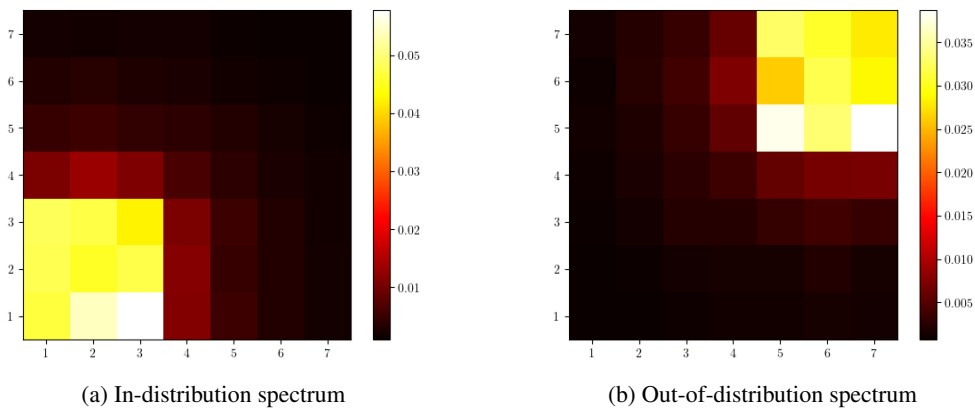

(a) In-distribution spectrum        (b) Out-of-distribution spectrum

Figure 6: Fourier spectra for in-distribution and out-of-distribution data in the ablation study

ple of a two-dimensional subgrid problem with resolution $64$ ($8 \times 8$). We can see that even though figure 1 shows that our model is around 10 times more accurate on average, both neural networks' relative error is still good enough to produce a visually similar solution for in-distribution data. On OOD data, however, it is visually clear that the fully connected neural network does not learn an accurate solution operator, while our model is able to adapt to the new distribution with high accuracy. The convolutional network fares in between, producing a similar solution but less accurate than our model. Figure 3 shows the same phenomenon in one dimension, in a subgrid problem with a resolution of $32$ (we note that all solutions are linearly interpolated back to the original grid size for comparison).

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

# A APPENDIX

## A.1 ERROR MEASURES

The relative errors plotted in figure 1 are calculated as the average normalized error of the predicted solutions. More precisely we calculate

$$\epsilon = \left( \frac{1}{N} \sum_{i=1}^{N} \frac{\| f_\theta(u_i(\boldsymbol{x}, 0)) - u_i(\boldsymbol{x}, t) \|^2}{\sigma_\rho^2} \right)^{\frac{1}{2}}$$

where $\sigma_\rho^2 = \mathbb{E} \left[ \int_0^T \int_{\boldsymbol{x}} (u(\boldsymbol{x}, t) - \bar{u}(\boldsymbol{x}, t))^2 \, d\boldsymbol{x} dt \right]$ is a normalization factor that sets the variance of the initial data (as a function of $x$) to be one, and allows us to do a fair comparison across distributions.

## A.2 TRAINING DYNAMICS

We can see in figure 7 that the training dynamics of our model are a lot smoother than those of the standard neural networks, leading to less volatility in the training and a more stable model. Furthermore, tables 1 and 2 show the number of parameters of our model and the standard neural networks as a function of the subgrid size. We can see that our model has fewer parameters in general, which is desirable for computational efficiency.

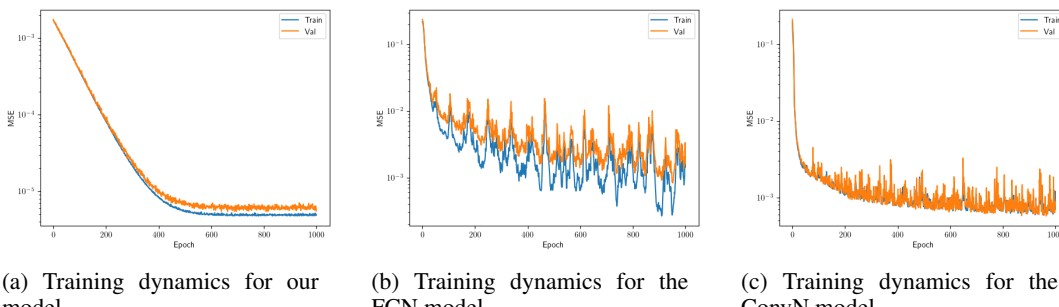

(a) Training dynamics for our model

(b) Training dynamics for the FCN model

(c) Training dynamics for the ConvN model

Figure 7: Training dynamics of our model and the standard neural networks in a two-dimensional subgrid of resolution $16 \times 16$

## A.3 MODEL ARCHITECTURE

Our model architecture is motivated by the forward Euler method for solving PDEs numerically. The core algorithm of the Euler method for solving the heat equation in one dimension is given by (Larsson & Thomée, 2009)

$$U_j^{n+1} = a(x)\lambda \left( U_{j+1}^n - 2U_j^n + U_{j-1}^n \right) + U_j^n \tag{4}$$

where $U_j^n = u(x_j, t_n)$ represents the solution at space point $j$ and time step $n$, $a(x)$ represents the non-constant coefficients of the equation, and $\lambda = \frac{dt}{dx^2}$.

We can then see that each layer in our model's architecture in equation 3 is built to resemble equation 4:

- $\lambda(U_{j+1}^n - 2U_j^n + U_{j-1}^n)$ is replaced by our fixed convolution operator $\text{conv}(W_{\text{Lap,dim}}, U)$.
- $a(x)$ is replaced by the bounded model parameters $\text{diag}(C_a \cdot \sigma(\theta))$.
- Adding this update back to $U_j^n$ in equation 4 is analogous to adding the update to $\boldsymbol{x}$ in equation 3.

At its core, our model is designed to learn a solution operator that resembles the Euler method but at coarser grids, with coarser (fewer) coefficients as the resolution decreases.

