# OpenReview forum: "Deep PDE Solvers for Subgrid Modelling and Out-of-Distribution Generalization"
_ICLR.cc/2024/Conference — Submitted to ICLR 2024_

### Official Review · Reviewer_QFWP · 2023-10-29

**Soundness:** 2 fair
**Presentation:** 2 fair
**Contribution:** 1 poor
**Rating:** 3
**Confidence:** 4

**Summary:**

The authors introduce an architecture to be used for building subgrid PDE solvers. The architecture embeds a discretized Laplacian operator in its convolution filters, which are multiplied by unknown coefficients. The authors apply the method in 1d and 2d linear heat equations and show that the learned model has some zero-shot capabilities for solutions with energy spectra not seen in training.

**Strengths:**

* The paper is generally well written and easy to follow.
* The proposed technique demonstrate some ability to adapt to data with different energy spectra than those encountered during training.

**Weaknesses:**

My most fundamental concerns with this work are mainly on the novelty and significance fronts
* The proposed framework does not show enough novelty. The main adaptation in the architecture is simply embedding a discretized Laplacian structure in the convolution filters in order to reduce the total number of parameters in the model.
* The problem setting is too simplistic to demonstrate significance. The authors exclusively focus on the linear heat equation with nonhomogeneous unknown coefficients, which is nowhere near the complexity of the other problems referenced, including Navier-Stokes and weather/climate. The heat equation has linear dynamics and generally admits smooth solutions, which means errors do not accumulate over time in complicated ways and solution has very little small-scaled features (so that representing the solution/dynamics fields is relatively easy). The claim that it is "complex enough" (page 4 top) is too much of a stretch. It is certainly not evident that the same OOD generalization might be observed on, for example, turbulent flows.

Less fundamental concerns:
* Baselines: FCN should not be considered a valid baseline if the dynamics is known to be local. In addition, the both FCN and ConvN show signs of overfitting, which suggests the architecture/training is sub-optimal. Techniques such as layer/group normalizations and learning rate annealing should at least be considered.

**Questions:**

Potential typo second line of page 7: "sample it up in space" - do you mean sample it down in time?

---

> ### Author Response · Authors · 2023-11-15
> **Clarifications on complexity of modelled problem**
>
> Thank you for your review and valuable feedback!
>
> We note that our paper is intended as a proof-of-concept architecture and we show a result which has not been demonstrated by other methods, which is Out-of-Distribution generalization. Furthermore, our paper shows results based on data sampled from different Fourier spectra, which is also an improvement over many other published papers where the distributions are essentially modelled by one parameter.
>
> What we meant when mentioning that the modelled problem is complex enough is that it is sufficient to highlight the results of our proof-of-concept architecture and shows that the proposed method works and is promising for CWM. Our goal is to show that this is a starting point to solve harder problems (like Navier Stokes) but we are not claiming to have solved them. We are currently working on extending the results to general advection-diffusion equations and the goal is to further extend it to all parabolic PDEs.
>
> Concerning the baselines used, we picked a standard fully connected net and a standard convolutional net as proxies for off-the-shelf ML models. We do agree though that a comparison with more complex baselines would be beneficial, and we aim to add more experiments to compare our model.
>
> As for the typo, it has been fixed, thanks for the catch! (we did mean "sample it down in time")

---

> > ### Comment · Reviewer_QFWP · 2023-11-21
> >
> > Thank you for the explanation.

---

### Official Review · Reviewer_6z1F · 2023-10-31

**Soundness:** 2 fair
**Presentation:** 2 fair
**Contribution:** 2 fair
**Rating:** 6
**Confidence:** 2

**Summary:**

This paper presents a PDE method to tackle the OOD problem of subgrid modeling. Here the new method is compared to a FCN and a CNN on a one and a two-dimensional dataset of the heat equation and the performance on OOD cases with respect to the Fourier spectrum.

**Strengths:**

Originality
First work tackling this problem

Significance
OOD generalization is an important problem in ML and CWM

**Weaknesses:**

- Not really clear what the ML part is here or if there even is one
- Not clear if this method scales to larger scale problem (which any relevant CWM problem would be)

**Questions:**

- Abstract should not include references
- Abstract does not say anything about the type of ML approach that is going to be used, e.g. architecture
- No conclusion section, please add, especially given that there is space left up to the page limit of 9 pages
- It is a little hard to see what the ML or even DL part is here. A graphic showing that would be nice.
- Not sure if Deep PDE solver is the right term here, there is not any Deep Learning used, if I understand correctly
- Why is OOD in the Fourier space? It would be nice to have an explanation about why this is the relevant OOD case

---

> ### Author Response · Authors · 2023-11-14
> **Clarifications on ML aspects of the paper**
>
> Thank you for your review and feedback, we are happy to clarify the points mentioned about our paper:
>
> - The model proposed is a deep neural network that is trained with ML algorithms to learn the parameters of the Model. The model architecture is explained in section 4 (specifically equation 3 and the paragraph above it). We appreciate the feedback and will consider adding a graphic for visualization.
>
> - The goal of our research is to show that the method does scale to larger-scale problems. In the paper, we show results for a specific dynamic (heat equation with non-constant coefficients) as a proof-of-concept that the proposed method works and is promising for CWM. We are currently working on extending the results to general advection-diffusion equations and the goal is to further extend it to all parabolic PDEs.
>
> - The architecture proposed is novel and it is based on expert knowledge of the structure of PDE solution operators, which is what is mentioned in the abstract. We have also removed the citations from the abstract.
>
> - We explain our results throughout the paper and focus on the main takeaways in the introduction. We did not opt for a conclusion section to avoid sounding repetitive.
>
> - There are many types of OOD generalization relevant to CWM and in our paper, we focus on initial data with a different shape from data previously seen by the models (as explained in section 3.3). This corresponds to the problem of having the same physical dynamics, but a different distribution of the density of particles (e.g. a more oscillatory density profile). Thus, to represent datasets with different shapes, we generate samples from different Fourier Spectra.
>
> Please let us know if you have further questions.

---

> > ### Comment · Reviewer_6z1F · 2023-11-18
> >
> > Thanks a lot for the clarifications!

---

### Official Review · Reviewer_wuBe · 2023-11-01

**Soundness:** 2 fair
**Presentation:** 2 fair
**Contribution:** 2 fair
**Rating:** 3
**Confidence:** 2

**Summary:**

A method is proposed for learning subgrid PDE solvers from gridded data examples. The approach uses a constrained convolutional neural network architecture. Experiments demonstrate the value of over simpler neural network approaches (fully connected, vanilla convolutional).

**Strengths:**

- The approach is simple and appears to work in experiments.
- The authors take care to consider the stability conditions.

**Weaknesses:**

- The paper was hard to follow.
- Details of the neural network training were missing, including optimizer, learning rate, stopping criteria, and hyperparameter optimization. In the appendix, the authors comment on the learning dynamics but its impossible for such a comparison to be meaningful without details of the learning algorithms.
- I would have liked to see more justification or an experimental ablation study to corroborate the statements in Section 4, including the statement that the identical layers ensures the same physical process between each physical time step and the coefficient bounds truly force the model to find a solution that is constrained physically and has the same benefits of traditional PDE solvers.

**Questions:**

- In the introduction, what does out of distribution data mean in this context?
- What is the use case? I understand that this is for situations where the PDE is unknown (Section 3.2), and the training data consists of one (or more?) gridded data example.
- If we don't know the PDE coeffients anyway, why not just coarsen the training data into the desired grid and learn the PDE coefficients directly?
- Rather than presenting the results figures right up front, it would be helpful to have more motivation for why this new approach is needed.

---

> ### Author Response · Authors · 2023-11-14
> **Clarifications on paper**
>
> Thank you for your review and feedback, we are happy to clarify the points mentioned about our paper:
>
> - We have attached the code as supplementary material including the details of the optimization. More specifically, all models were trained with the Adam optimizer for 1000 epochs and no weight decay. We have ensured that the training parameters are equal for all models to ensure fair comparisons, but the optimization itself is not the objective of our paper.
>
> - The composition of identical layers ensures by construction that the same process is applied at each time step. Similarly, by constraining the coefficients to the stability bounds of the PDE, we ensure by construction that the model must learn a solution in the stability region of the PDE. We note that if the bounds are not included, the model might converge to a sub-optimal solution that lies outside the stability region of the PDE. We will consider adding a section showing this in the appendix for more clarity on its importance.
>
> - We explain in the last paragraph of section 3.3 the meaning of Out-of-Distribution data: "Here we consider OOD to be initial data with a different shape (Fourier spectrum) from data previously seen by the models. This corresponds to the problem of having the same physical dynamics, but a different distribution of the density of particles (e.g. a more oscillatory density profile)."
>
> - Correct, the use of our approach is to learn a solution operator for an unknown PDE given observations of solutions in a given coarse grid. What we show is that having some prior knowledge of the physical process behind it (in this case knowing it behaves like a heat equation but we don't know the coefficients) can help us engineer an architecture that produces a better model than standard ML approaches. We are currently working on extending our results to general advection-diffusion equations and the goal is to further extend it to all parabolic PDEs.
>
> - In reality, we will not have access to fine-grid training data, so we are more interested in having a model that learns a solution operator rather than coefficients. We note that while we did observe that the coefficients learnt were a  (non-linear) average of the ground truth coefficients, this might not be the case when we extend our work to more complex dynamics.
>
> - Thank you for the feedback, we will consider explaining more about the motivation behind our research.
>
> Please let us know if you have further questions.

---

### Official Review · Reviewer_Ap8L · 2023-11-03

**Soundness:** 2 fair
**Presentation:** 2 fair
**Contribution:** 1 poor
**Rating:** 5
**Confidence:** 4

**Summary:**

The paper introduce a method to address the subgrid problems: given training trajectory on the find grid, learn a solution operator on coarser grids. Based on four principles: locality, stability, linearity, memory-less, the authors introduce an architecture that respects these principles. Experiments show that the method outperforms FCN and CNNs on both in-distribution and out-of-distribution tasks of heat equation with non-constant coefficients.

**Strengths:**

Significance: the authors address an important problem in the domain. Also, out-of-distribution generalization is important for neural PDE methods.

Novelty and contribution: the authors develops neural architectures that respect several important physics principles. As is generally known, incorporating more physics knowledge can improve the model's generalization and data-efficiency. The method is to be commended for its consideration of important physics principles.

Clarity: the paper is mostly clear.

**Weaknesses:**

Generality: one concern I have for the paper is generality. The paper addresses a very narrow problem: heat equation, which is linear, and has specific structures. Thus, the proposed architecture is linear and has specific convolutional kernels, and cannot generalized to other equations. The improvements in accuracy is purely due to the specific physics priors embedded into the architecture, which is to be expected. Overall, the method in its current form lacks generality as a general subgrid method. This aspect could be improved, e.g., by considering a more general approach (or meta-method) that can build physics priors into more general kinds of equations.

Soundness: the paper lacks comparison with some important baselines, e.g., Fourier Neural Operators and U-Net. It would be great if the authors performs experiments on these method, to demonstrate the effectiveness of the proposed method.

**Questions:**

N/A

---

> ### Author Response · Authors · 2023-11-14
> **Paper scope and work in progress**
>
> Thank you for your comments and valuable feedback!
>
> We agree that the scope of the paper is narrow and we address only a very specific problem. The work presented is a starting point for the general subgrid problem and is intended as a proof-of-concept architecture. We are currently working on extending the results to general advection-diffusion equations and the goal is to further extend it to all parabolic PDEs. We also agree that a comparison with current state-of-the-art PDE solvers would be beneficial, and we aim to add experiments comparing our model with those baselines.
>
> As mentioned, our paper is intended as a proof-of-concept architecture and we show a result which has not been demonstrated by other methods, which is Out of Distribution generalization. Furthermore, our paper shows results based on data sampled from different Fourier spectra, which is also an improvement over many other published papers where the distributions are essentially modelled by one parameter.

---

> > ### Comment · Reviewer_Ap8L · 2023-11-23
> > **Official Comment**
> >
> > Thanks for the explanation. I agree that the paper is a very good starting point. I will remain my score, and I encourage the authors to continue the work, which could result in a strong paper in the future.

---

### Meta-Review · Area_Chair_MYK2 · 2023-12-10

**Metareview:**

This paper introduces a neural architecture for building sub-grid partial differential equations (PDE) solvers, with a discretised Laplacian operator embedded within the convolution filters. The method is applied to learn to integrate 1D and 2D linear heat equations and can generalised to solutions with out-of-distribution energy spectra.

Strengths:
* Reviewers (Ap8L,6z1F) praised addressing an important problem in the domain and incorporating physics priors.
* Reviewer wuBe praised the care in considering stability conditions of the equations.
* Out-of-distribution generalisation (QFWP,6z1F)

Weaknesses:
* Lack of clarity in the paper (wuBe,6z1F)
* The paper addressed the very narrow problem of linear heat equation (Ap8L) that may not scale to climate or weather (6z1F,QFWP) and Navier-Stokes in general (QFWP)
* Physics priors embedded in the architecture lack generality as a subgrid method (Ap8L)
* Lack of comparison with baselines such as Fourier Neural Operators or UNets (Ap8L)
* Missing details about the neural network architecture and training (wuBe)
* Missing ablations to prove that the solutions found are constrained physically (wuBe)

Based on the scores (3, 3, 6, 6), the paper does not meet the bar for acceptance.
As the authors recognise, the paper is a proof of concept and they are currently working on extending the paper to more general advection-diffusion equations, and I encourage the authors to submit it to a workshop while working on extending results towards a larger paper.

**Justification For Why Not Higher Score:**

The reviewers found the paper to show promising but still preliminary results in PDE solving.

**Justification For Why Not Lower Score:**

N/A

---

### Decision · Program_Chairs · 2024-01-16

Reject